# Shared Molecular Genetic Mechanisms Underlie Endometriosis and Migraine Comorbidity

**DOI:** 10.3390/genes11030268

**Published:** 2020-02-29

**Authors:** Emmanuel O. Adewuyi, Yadav Sapkota, Asa Auta, Kosuke Yoshihara, Mette Nyegaard, Lyn R. Griffiths, Grant W. Montgomery, Daniel I. Chasman, Dale R. Nyholt

**Affiliations:** 1School of Biomedical Sciences, Faculty of Health, and Institute of Health and Biomedical Innovation, Queensland University of Technology, Brisbane, Queensland 4000, Australia; lyn.griffiths@qut.edu.au; 2Department of Epidemiology and Cancer Control, St. Jude Children’s Research Hospital, Memphis, Tennessee 38105, USA; yadav.sapkota@stjude.org; 323andMe, Inc., 899 W. Evelyn Avenue, Mountain View, California 94041, USA; publication-review@23andMe.com; 4School of Pharmacy and Biomedical Sciences, University of Central Lancashire, Preston PR1 2HE, UK; aauta@uclan.ac.uk; 5Department of Obstetrics and Gynecology, Niigata University Graduate School of Medical and Dental Sciences, Niigata 950-2181, Japan; yoshikou@med.niigata-u.ac.jp; 6Department of Biomedicine – Human Genetics, Aarhus University, DK-8000 Aarhus, Denmark; nyegaard@biomed.au.dk; 7iPSYCH, The Lundbeck Foundation Initiative for Integrative Psychiatric Research, DK-2100 Copenhagen, Denmark; 8Institute for Molecular Bioscience, The University of Queensland, Brisbane, Queensland 4072, Australia; g.montgomery1@uq.edu.au; 9Divisions of Preventive Medicine, Department of Medicine, Brigham and Women’s Hospital, Harvard Medical School, Boston, MA, USA; DCHASMAN@research.bwh.harvard.edu

**Keywords:** causality, comorbidity, endometriosis, gene-based association study, genetic overlap, GWAS, Mendelian randomisation, migraine, molecular genetics, pathway enrichment study

## Abstract

Observational epidemiological studies indicate that endometriosis and migraine co-occur within individuals more than expected by chance. However, the aetiology and biological mechanisms underlying their comorbidity remain unknown. Here we examined the relationship between endometriosis and migraine using genome-wide association study (GWAS) data. Single nucleotide polymorphism (SNP) effect concordance analysis found a significant concordance of SNP risk effects across endometriosis and migraine GWAS. Linkage disequilibrium score regression analysis found a positive and highly significant genetic correlation (*r*_G_ = 0.38, *P* = 2.30 × 10^−25^) between endometriosis and migraine. A meta-analysis of endometriosis and migraine GWAS data did not reveal novel genome-wide significant SNPs, and Mendelian randomisation analysis found no evidence for a causal relationship between the two traits. However, gene-based analyses identified two novel loci for migraine. Also, we found significant enrichment of genes nominally associated (*P*_gene_ < 0.05) with both traits (*P*_binomial-test_ = 9.83 × 10^−6^). Combining gene-based *p*-values across endometriosis and migraine, three genes, two (*TRIM32 and SLC35G6)* of which are at novel loci, were genome-wide significant. Genes having *P*_gene_ < 0.1 for both endometriosis and migraine (*P*_binomial-test_ = 1.85 ×10^−^°^3^) were significantly enriched for biological pathways, including interleukin-1 receptor binding, focal adhesion-PI3K-Akt-mTOR-signaling, MAPK and TNF-α signalling. Our findings further confirm the comorbidity of endometriosis and migraine and indicate a non-causal relationship between the two traits, with shared genetically-controlled biological mechanisms underlying the co-occurrence of the two disorders.

## 1. Introduction

Endometriosis is one of the leading gynaecological disorders affecting 6%–10% of women of reproductive age and 35%–50% of women with infertility worldwide [1,2]. The disorder is defined by the presence of endometrial tissue in extra-uterine locations and characterised by varying degrees of pelvic, menstrual, abdominal, bowel and lower-back pain as well as infertility [1,2]. With an estimated global prevalence of 14.7%, migraine, on the other hand, is the most disabling neurologic disorder and the third most common illness worldwide [3,4]. Like endometriosis, women in their reproductive and most productive years are more commonly affected with migraine [5,6]. A typical migraine presents with a recurrent, unilateral and episodic headache of moderate to severe intensity [7]. Both endometriosis and migraine portend substantial morbidity with wide-ranging socioeconomic burdens to sufferers, their families, relationships, and the society at large [8,9,10,11]. Notably, the diagnosis of the two disorders is challenging, due to a lack of diagnostic markers, which often results in missed or delayed diagnosis. Also, the aetiology and pathogenesis of endometriosis and migraine remain relatively obscure, and there are currently no known curative treatments for them.

While endometriosis and migraine appear to have clear-cut distinctions—anatomically, as well as in terms of clinical diagnosis and disease classification—some shared epidemiological characteristics or similarities suggest a comorbid relationship between them. For instance, similar to endometriosis, which almost exclusively affects women [12], migraine has a substantially higher prevalence in women (15%‒18%) compared to men (6%), and women of reproductive age also experience a longer duration of migraine attacks with greater disability [13,14,15,16]. The two disorders share similar risk factors in women including early menarche, menorrhagia, and involvement of the menstrual cycle in their pathogenesis [17,18,19]. Indeed, increased exposure to menstruation is a known risk factor for endometriosis just as menstrual migraine and menstrually-related migraine (with prevalence varying from 4%–70%) are common subtypes of migraine in women [20,21,22,23,24]. Furthermore, Danazol (a synthetic androgen for managing endometriosis) has been reported to reduce the frequency of migraine attacks [25]. In addition to their shared similarities and risk factors, the comorbidity (co-occurrence of two or more conditions in the same individual) of endometriosis with migraine has been consistently reported by observational epidemiological studies [5,6,18,26,27,28,29]. 

As far back as 1975, for example, a clinic-based study had reported a higher prevalence of headache (84%) among women diagnosed with endometriosis compared to the control (60%, *P* = 0.007) [26]. Interestingly, 28% of the endometriosis cases described their headaches as migraine-like compared to only 18% (*P* = 0.023) in control [26]. In a related case-control study, over two times higher prevalence of migraine was found in endometriosis (38.3%) compared to the control (15.1%, *P* < 0.001) [27]. A study investigating the cost implications of endometriosis in the United States similarly found a three-fold greater prevalence of migraine in endometriosis compared to the general population [28]. More recently, adolescents with surgically confirmed endometriosis had over two-fold greater prevalence (69.3% vs 30.7%) and nearly five-fold increased odds of migraine (adjusted Odds Ratio [AOR] = 4.77; 95% CI: 2.53–9.02) compared to their counterparts with no endometriosis [30]. Also, a recent French case-control study similarly found a higher prevalence of endometriosis in migraine cases (35.2%) compared to controls (17.4%, *P* = 0.003) [5].

The consistent and growing evidence on the endometriosis–migraine comorbid relationship, notwithstanding, some questions remain unanswered. First, is the endometriosis–migraine comorbidity reported in observational studies a true association, or could the findings be due to the confounding effects, biases, or otherwise false-positive results of the traditional observational studies? Second, is there a causal relationship between endometriosis and migraine? Third, are there some shared genetic variants, susceptibility loci, genes and biological pathways between the two disorders? Last, what biological mechanism(s) may underlie possible endometriosis–migraine comorbidity? 

Using a twin-based study approach, Nyholt et al. [13] examined the genetic influences and comorbidity of migraine and endometriosis and reported that additive genetic factors accounting for 69% (95% CI: 60%–77%) of the phenotypic variance in migraine also account for 17% (95% CI: 8%–27%) of the variance in endometriosis (i.e., bivariate heritability of 17%)—suggesting shared genetic influences completely explain their co-occurrence within individuals. Additional bivariate heritability analyses utilising direction-of-causation twin models did not support endometriosis as the cause of migraine or vice versa; however, given the sample size and similar heritability for endometriosis and migraine, these analyses lacked power [13].

To date, genome-wide association studies (GWAS) have identified 19 independent single nucleotide polymorphisms (SNPs) for endometriosis [31] and 44 for migraine [32]. However, molecular genetic studies of the association between endometriosis and migraine, including causality and shared genetic risk variants and loci are currently lacking. Lastly, biological pathways driving possible endometriosis and migraine comorbidity remain poorly understood. The present study, thus, aims to assess the molecular genetic overlap, causal relationship and shared pathways between endometriosis and migraine using GWAS data.

## 2. Materials and Methods 

### 2.1. Data Sources and Study Samples

We utilise GWAS meta-analysis summary statistics from the International Endogene Consortium (IEC, endometriosis GWAS data) [31] and the International Headache Genetics Consortium (IHGC, migraine GWAS data) for analysis in the present study. Summary statistics data sourced from the United Kingdom Biobank (UKBB, migraine GWAS data) were used in testing the reproducibility of our findings for SNP-level genetic overlap and correlation studies.

#### 2.1.1. IEC Endometriosis GWAS Data

The ‘IEC endometriosis’ GWAS summary statistics utilised in this study represent the largest endometriosis genetic study published to date [31]. The data combined 11 separate GWAS case-control datasets (QIMRHCS, deCODE, LEUVEN, OX, 23andMe, NHS2-dbGaP, WGHS, iPSYCH, BBJ, Adachi-6, and Adachi-500K) consisting of 17,054 cases of endometriosis (all stages of endometriosis) and 191,858 controls (n = 208,912). A total of 6,979,035 SNPs passed quality control in six or more (at least 50%) of the studies and those were included in a fixed-effect meta-analysis [31]. Study participants in the GWAS were of European (approximately 93%) and Japanese ancestries (from Australia, Belgium, Denmark, Iceland, Japan, the UK, and the USA). Endometriosis was surgically confirmed (using the revised American Fertility Society system [33]) in cases from QIMRHCS, OX, deCODE and LEUVEN studies, while cases from other studies were self-reported or their diagnosis was based on combined self-report and surgical records [31]. Similar quality control procedures were used in each of the GWAS. A detailed description of these GWAS, the quality control and the analyses carried out have previously been published [31]. 

#### 2.1.2. IHGC Migraine GWAS Data

Our migraine data were sourced from the 2016 IHGC (http://www.headachegenetics.org) migraine GWAS, which meta-analysed migraine summary statistics from 22 GWAS (obtained from six tertiary headache clinics and 27 population-based cohorts) [32]. A total of 59,674 migraine cases and 316,078 controls were included in the meta-analysis, and all participants were unrelated individuals of European ancestry [32]. Diagnosis of migraine was through self-reported questionnaires or clinical interview, and, in line with the criteria of the International Classification of Headache Disorder (ICHD) [34]. Standard protocols for quality control were included and a common 1000 Genomes Project [35] reference panel (Phase I, v3) was used in imputing missing genotypes into each of the samples. Logistic regression analysis was conducted on the imputed genotypes in each of the GWAS for association analysis [32]. 

To account for possible population stratification and other confounders, an adjustment was made for the top ten principal components, sex and other covariates where necessary [32]. The GWAMA program [36] was used to perform a combined fixed-effect meta-analysis. SNPs were filtered based on imputation quality and other metrics [32]. A detailed and more comprehensive description of the ‘IHGC migraine’ GWAS sample has previously been published [32]. The data utilised in the present study were restricted to 29,208 cases and 172,931 controls (*n* = 202,139) with a total of 8,935,979 SNPs, following the exclusion of the 23andMe GWAS sample (30,465 migraine cases and 143,147 controls). The 23andMe GWAS sample was excluded to ensure there was no sample overlap between the ‘IEC endometriosis’ GWAS data (which comprise 23andMe GWAS data) and the ‘IHGC migraine’ GWAS data.

#### 2.1.3. United Kingdom (UK) Biobank Data

The UK Biobank is a large, population-based cohort study that was established in the United Kingdom in the year 2006. A total of 500,000 volunteers aged 40–69 years were recruited for the study between 2006 and 2010 with the aim of investigating the genetic and environmental determinants of health and diseases [37]. Extensive genotype and phenotype data, including biological samples, physical measurements, health and lifestyle information, multimodal imaging, and genome-wide genotyping have been collected from these study participants [37]. Also, a variety of their health-related outcomes are being followed up [37]. Anonymised data from the study are made available to researchers via an application process [37]. We utilised UK Biobank GWAS summary statistics for migraine sourced from the Neale Lab, which performed linear regression analysis controlling for 10 principal components of ancestry, in a sample of 337,159 unrelated individuals of “White British” ancestry, comprising 10,007 self-reported migraine cases and 327,152 controls (https://nealelab.github.io/UKBB_ldsc/h2_summary_20002_1265.html, downloaded 12/03/2018). GWAS summary statistics were available for 10,894,597 SNPs.

### 2.2. SNP Effect Concordance Analysis (SECA)

We assessed the genetic overlap between the ‘IEC endometriosis’ GWAS data and the ‘IHGC migraine’ GWAS data using SECA (https://sites.google.com/site/qutsgel/software/seca-local-version) [38]. SECA utilises GWAS summary statistics data and tests whether the direction of single nucleotide polymorphism (SNPs) are positively correlated across GWAS results thereby facilitating the assessment of genetic overlap between traits [38]. We formatted our datasets appropriately so that SECA requirements were met [38]. Thereafter, the ‘IEC endometriosis’ GWAS data was assigned, for SECA analysis, as dataset 1 and the ‘IHGC migraine’ GWAS as dataset 2. SECA first aligns the SNP effects across dataset 1 and dataset 2 to the same effect allele, and, subsequently extracts a subset of independent SNPs by utilising a ‘*p*-value informed’ SNP clumping, accounting for linkage disequilibrium (LD) between SNPs. 

For each of the ‘IEC endometriosis’ and ‘IHGC migraine’ GWAS datasets, SECA partitions the extracted SNPs into 12 *p*-value subsets which ranges from 0.01 to 1 (*P* ≤ 0.01, 0.05, 0.1, 0.2, 0.3. 0.4, 0.5, 0.6, 0.7, 0.8, 0.9, 1.0). The *p*-value partitioning yields 144 subsets of SNPs from all possible combinations of dataset 1 (*P1*, 12 SNP subsets) with dataset 2 (*P2*, 12 SNP subsets). SECA performs two tests: a binomial test to assess the presence of excess SNP subsets associated between the two datasets, and, the Fisher exact test for the concordance in the direction of effect of the individual SNPs across datasets 1 and dataset 2 [38]. 

Our SECA analysis was restricted to SNPs that are most strongly associated with dataset 1; hence, we swapped ‘IEC endometriosis’ GWAS data as dataset 2 and ‘IHGC migraine’ GWAS data as dataset 1 in an analogous analysis. This ability to condition on one of the GWAS datasets (not possible using the linkage disequilibrium score regression method can help determine whether an observed genetic overlap is driven similarly by both datasets, or driven predominantly by one dataset. We estimated LD using the 1000G Phase I v3 CEU genotype data and LD pruning prioritised SNPs with smaller *p*-values (*P1*) in dataset 1. Also, we tested the reproducibility of our study using independent migraine summary statistics GWAS data from the UKBiobank.

### 2.3. Linkage Disequilibrium Score Regression (LDSC)

We estimated the SNP-based heritability and cross-trait genetic correlation for endometriosis and migraine using the LDSC software (https://github.com/bulik/ldsc). The ‘IEC endometriosis’ and ‘IHGC migraine’ GWAS data were utilised in the analysis. These datasets were formatted using the ‘munge_sumstats.py’ script in line with the LDSC documentation (https://github.com/bulik/ldsc/wiki/Heritability-and-Genetic-Correlation). We performed univariate LDSC analyses to estimate SNP-based liability heritability (*h*^2^_SNP_) using the ‘IEC endometriosis’ (sample prevalence = 8.2%, population prevalence = 8% [31]) and the ‘IHGC migraine’ (sample prevalence = 14.5%, population prevalence = 15% [39])’ GWAS data. Also, to estimate the genetic correlation (*r*_G_) between the two traits, we conducted a bivariate cross-trait LDSC analysis utilising the ‘IEC endometriosis’ GWAS data and the ‘IHGC migraine’ GWAS data. This analysis complements our SECA-based study in assessing the genetic overlap between endometriosis and migraine. We constrained the intercept to one for ‘IEC endometriosis’ GWAS data (both in heritability and cross-trait LDSC correlation analysis) because the estimated intercept (without constraining) was not significantly different from one. We also constrained the genetic covariance intercept to zero (in the cross-trait LDSC correlation analysis) given there was no sample overlap between the two datasets. The intercept for all migraine data was significantly different from one, hence, their estimated intercepts (obtained without constraining) were retained in the model. In all the LDSC analyses, we calculated the LD scores based on the European 1000 Genomes Project haplotype reference data (Phase I, v3). Last, we repeated the above analysis procedures using the ‘IEC endometriosis’ GWAS data and the migraine GWAS data from the UKBB (‘UKBB migraine’ GWAS data). 

### 2.4. Cross-Disorder Meta-Analysis of Endometriosis and Migraine

We conducted a cross-disorder meta-analysis of the ‘IEC endometriosis’ and the ‘IHGC migraine’ GWAS summary statistics data to identify possible genetic variants and loci shared by both endometriosis and migraine. The inverse variance-weighted fixed effect (FE) and ‘Han and Eskin’s random effect’ (RE2) models, implemented in METASOFT (http://genetics.cs.ucla.edu/meta/), were utilised in the meta-analysis. We accounted for possible between-study heterogeneity using RE2—a modified random effect model. Unlike the traditional random effect (RE) model, which is highly conservative, RE2 has greater power under heterogeneity [39,40,41]. We included a total of 411,051 participants in the analysis, and meta-analysed the 6,904,914 SNPs overlapping the two GWAS. We aimed at identifying novel cross-disorder genome-wide significantly enriched (*P* < 5 × 10^−8)^) SNPs and loci associated with both endometriosis and migraine.

### 2.5. Mendelian Randomisation (MR)

To assess the causal relationship between endometriosis and migraine, we performed a two-sample Mendelian Randomization analysis (“TwoSampleMR”) [42] utilising genome-wide significant (*P* < 5 × 10^−8^) SNPs associated with ‘IEC endometriosis’ summary statistics data. Randomised controlled trials (RCTs) are considered the most reliable evidence for drawing causal inferences. However, due to limitations such as substantial costs, non-availability of appropriate interventions and controls and certain ethical constraints [43,44], conducting an RCT may not always be feasible. MR analysis mimics the design of an RCT thereby providing an alternative approach to assessing and estimating the causal relationship between an exposure and outcome variables [45]. 

MR method is anchored on the principle of Mendel’s law of inheritance—gene segregation and natural randomisation at gamete formation which is comparable to the experimental randomisation in RCTs. The method is supported by the understanding that genotypes are naturally fixed at conception and generally not subject to confounding effects or bias of reverse causation [45]. MR analysis, thus, exploits the presence of specific genetic variants associated with the variable of interest as proxies for assessing causality with the outcome of interest. The effect of the genetic variants (instrumental variables, IVs) on the outcome is expected to be through the exposure variable (vertical pleiotropy). Although not without limitations—possible violations of some of its assumptions—MR analysis is increasingly being used as an unbiased causality detection, and, where possible, estimation method [45].

In the present study, we performed “TwoSampleMR” analyses [42]. First, we extracted a total of 338 SNPs associated with endometriosis, in the ‘IEC endometriosis’ GWAS data, at a genome-wide significance level (*P* < 5 × 10^−8^). We assigned endometriosis as the exposure variable and migraine (‘IHGC migraine’ GWAS data) as the outcome variable. Following LD clumping (*r*^2^ < 0.001; to ensure the independence of the extracted SNPs), 11 genome-wide significant SNPs associated in the ‘IEC endometriosis’ GWAS were retained as our IVs. Second, we extracted SNP effects from the outcome (‘IHGC migraine’ GWAS) data. To ensure that the SNP effects on exposure and outcome data correspond to the same allele, we carried out harmonisation of both the exposure and the outcome variables. 

Last, we conducted a “TwoSampleMR” analysis using the inverse variance weighted (IVW) method. IVW estimates are essentially the weighted average of the individual Wald-type ratios for each of the IVs. The IVW method assumes the absence of horizontal pleiotropy or a balance of same among the IVs. We conducted sensitivity analyses to address a possible violation of this assumption using the weighted median (which provides valid causal estimates even if up to 50% of the IVs have pleiotropic effect) [46], and the MR-Egger method (which corrects pleiotropy and provides valid causal estimates even if all the IVs are invalid) [47]. We implemented the “TwoSampleMR” [42] analysis methods in the R statistical package following a well-established protocol (https://mrcieu.github.io/TwoSampleMR/). 

MR analyses are based on three fundamental assumptions [48]. First, is that a robust association exists between the selected genetic variants (IVs) and the exposure variable [48]. This assumption can easily be validated, and we utilised only the genome-wide significant (*P* < 5 × 10^−8^) SNPs associated with endometriosis thereby satisfying the assumption. Second, is that the IVs are not associated with potential confounders [48]. We acknowledge that this assumption is difficult to prove, however, to reduce chances of violating it, we ensured that our IVs were independent. Also, we assessed the association between the IVs and age at menarche, age at menopause, menorrhagia, oestrogen level, as well as oral contraceptives use—all of which are possible risk factors for endometriosis and migraine [17,18,19]. This assessment was carried out using PhenoScanner v2 [49] (http://www.phenoscanner.medschl.cam.ac.uk, accessed on 2nd September 2019), at *P* < 1 × 10^−05^ (*suggestive* genome-wide significance level). Our IVs were not associated with any of these traits, except “rs74485684” which we found to be associated with ‘length of menstrual cycle’ and ‘excessive, frequent and irregular menstruation’ (Appendix A). To address possible pleiotropy implied by this finding, we carried out a ‘leave-one-out’ MR analysis.

The third assumption, which is also difficult to validate, is that the IVs do not affect the outcome through any alternative pathway other than the exposure variable, that is, there is no horizontal pleiotropy [48]. We conducted a test for horizontal pleiotropy as well as used alternative MR approaches including MR-Egger, and weighted median, to minimise the possibility of breaching this assumption. 

### 2.6. Gene-Based Association Study

Gene-based analysis examines associations between a trait of interest and all SNPs while accounting for LD and allelic heterogeneity between the SNPs [50]. Compared to SNP-level studies (in which individual SNPs are assessed), gene-based studies are more powerful in gaining mechanistic insights into the biology of complex traits [50] given that, as the basic functional units of the human genome, they are more closely related to biological mechanisms than SNPs. Thus, to identify genes associated with endometriosis and migraine as well as further assess the molecular genetic overlap between the two traits, we conducted gene-based tests using Vegas2 software (https://vegas2.qimrberghofer.edu.au/) [51]. Vegas2 is user-friendly, computationally tractable, and has been used extensively in studies [31,51]. We utilised the ‘IEC endometriosis’ and ‘IHGC migraine’ GWAS data for the Vegas2 gene-based analyses. 

Prior to conducting Vegas2 analyses, we extracted all SNPs from each of the two GWAS data. Following the exclusion of SNPs with no rsIDs, a total of 6,978,534 SNPs from the ‘IEC endometriosis’ GWAS and 8,175,736 SNPs from the ‘IHGC migraine’ GWAS were available for analysis. However, to ensure equivalent gene-based tests were performed for both disorders, we restricted Vegas2 analyses to a total of 6,904,914 SNPs overlapping the ‘IEC endometriosis’ and ‘IHGC migraine’ GWAS. We utilised the following Vegas2 options: Use SNPs from = ‘1000G EUROPEAN’; Select Sub-population from = ‘ALL EUROPEAN’; Use Gene definition from = ‘+/− 0 kb outside gene’; and Chromosome = ‘All’. Importantly, rather than the default ‘Top-x% test with top 100 per cent’ test, we specified the ‘Best-SNP test’. These analysis procedures were carried out separately for the ‘IEC endometriosis’ and ‘IHGC migraine’ GWAS data. We extracted nominally significant genes (at *P* < 0.1, *P* < 0.05, and *P* < 0.01) from Vegas2 outputs for each of the two traits and assessed those for overlapping genes between endometriosis and migraine. We also estimated gene-based Fisher’s combined *p*-values (FCP) for association (at *P*_gene_ < 0.1) across endometriosis and migraine to assess genes overlapping the two traits at a genome-wide level of significance.

Due to the presence of ‘LD between the most significant SNP (‘Best-SNP’) assigned to each gene, gene-based association results could be correlated across neighbouring genes’ [52]. Hence, we estimated the effective number of independent genes (independent gene-based tests) by examining the LD between the ‘Best-SNP’ assigned to each gene. Briefly, we estimated the effective number of independent gene-based tests in both the endometriosis and migraine datasets utilising the ‘genetic type 1 error calculator’ (GEC) software [53]. This analysis adjusts for multiple testing corrections taking into account correlation due to LD which may exist across neighbouring genes in our gene-based results. ‘Best-SNPs’ from the endometriosis and migraine Vegas2 results were processed as input files for GEC analysis [53]. GEC first partitions input SNPs into LD blocks with the assumption that LD blocks are independent (*r*^2^_LD_ < 0.1), and thereafter estimates the effective number of independent SNPs (hence, the independent gene-based tests) in the LD blocks.

### 2.7. Overlapping Genes and Statistics Tests

To allow for differences in power across the endometriosis and migraine GWA studies, we generated gene sets with gene-based association *p*-values less than three nominal *p*-value thresholds (*P_gene_* < 0.1, *P_gene_* < 0.05, and *P_gene_* < 0.01). For each gene set, estimates of the effective number of independent gene-based tests were calculated by GEC [53]. We assigned endometriosis as the ‘discovery’ set and migraine as the ‘target’ set to test whether the proportion of overlapping genes was more than expected by chance. The observed number of overlapping genes was defined as ‘the effective number of independent genes with *p*-values less than the threshold in both the discovery and target sets’ [52]. The observed proportion of overlapping genes was ‘calculated as the observed effective number of independent overlapping genes divided by the effective number of independent genes with a *p*-value less than the threshold in the discovery set’ [52]. The expected proportion of overlapping genes was calculated as the effective number of independent genes with a *p*-value less than the threshold in the target set divided by the total effective number of independent genes in the target set. The statistical significance of whether the number of overlapping genes was more than expected by chance was calculated using one-sided exact binomial test. We also report the raw number of genes in the gene sets to highlight the importance of estimating the effective number of independent genes.

### 2.8. Pathway-Based Functional Enrichment Analyses 

To further elucidate potential biological mechanisms underlying the co-occurrence of endometriosis and migraine, we conducted pathway-based functional enrichment analyses. The protocols proposed by Reimand and colleagues [54] for enrichment analysis (using the g:Gost tool in g:Profiler [55]), visualisation (using Enrichmentmap [56]) and interpretation (using auto annotate [54]) of enriched pathways were adopted in this study. The g:Gost tool, implemented in g:Profiler, performs statistical enrichment analysis and automates the functional annotation of user-inputted genes based on their molecular, cellular and biological functions [54,55], thereby identifying over-represented (significantly enriched) biological pathways for the trait(s) of interest [54,55]. We utilised the web-based (http://biit.cs.ut.ee/gprofiler/) version of the tool, which is user-friendly. Notably, g:Gost’s databases, including Gene Ontology, WikiPathways and Human Phenotype Ontology (for human disease phenotypes), are updated on a regular basis [55]. Regulatory motifs matches (TRANSFAC), miRNA targets (miRTarBase), Human Protein Atlas (for tissue specificity), CORUM (for protein complexes) and Biological pathways (Kyoto Encyclopedia of Genes [KEGG], as well as Reactome) are also included in the g:Gost tool of g:Profiler [55].

In the present study, we utilised the g:Gost tool of g:Profiler (accessed 1st October 2019) to perform pathway-based functional enrichment analysis [54,55] using genes overlapping endometriosis and migraine at *P*_gene_ < 0.1 [52]. We employed the ‘g:SCS algorithm’ recommended for multiple testing correction in the g:Gost analysis, and restricted our results to only significantly enriched pathways at *P*_adj_ < 0.05 (adjusted *p*-value for multiple testing correction [54]). Also, the size of the functional category (term size) was restricted to within 5 and 350 values (minimum and maximum) as recommended [54]. Several of the pathways enriched in g:Gost tool may be redundant. Therefore, we utilised the ‘Enrichmentmap’ application to produce ‘enrichment maps’ by collapsing related versions of over-represented pathways (g:Gost results) into simplified biological themes—thus, eliminating redundancy and enhancing the visualisation of enriched pathways [54,56]. We also utilised the ‘auto annotate’ application, to promote the interpretation of enriched pathways by organising ‘enrichment maps’ into clusters [54]. We implemented both ‘Enrichmentmap’ and ‘auto annotate’ tools via the Cytoscape environment [54,57].

## 3. Results

### 3.1. SECA: Genetic Overlap between Endometriosis and Migraine

SECA reveals significant concordance of SNP effects across the endometriosis and migraine GWAS, indicating that a strong molecular genetic overlap exists between the two traits. All 144 SNP subsets produced Fisher’s exact tests with at least nominally significant concordance effects (OR > 1 and *P* ≤ 0.05) between the ‘IEC endometriosis’ GWAS data (dataset 1) and ‘IHGC migraine’ GWAS data (dataset 2) [*P*_Fsig-permuted_ = 9.99 × 10^−04^; 95%CI: 5.12 × 10^−05^ – 5.64 × 10^−03^]. The most statistically significant concordance test was produced by SNP subsets with ‘IEC endometriosis’ GWAS *P*_assoc_ ≤ 0.2 and the ‘IHGC migraine’ GWAS *P*_assoc_ ≤ 0.6 (OR_FT_ = 1.36; *P*_FTmin-permuted_ = 1.66 × 10^−32^). Moreover, a total of 59,188 independent SNPs was shared by both endometriosis and migraine (SNP subsets with *P1* = *P2* = 1), out of which 30,790 (52%) showed concordance effect (Table 1). The test of association between the two traits (for SNP subsets with *P1* = *P2* = 1) was positive and highly significant statistically (OR = 1.18, Fisher’s *p*-value [two-sided] = 8.77 × 10^−23^). Importantly, SNP effect concordance increased as one conditioned on SNPs with smaller *p*-values. For example, the risk increasing alleles were concordant (OR = 1.92, Fisher’s *p*-value [two-sided] = 2.23 × 10^−09^) for 804 (58%) of the 1,383 independent endometriosis and migraine SNPs with nominally significant *p*-values (*P1* = *P2* < 0.05) [Table 1]. The proportion of concordance further increased to 66% (OR = 3.61, Fisher’s *p*-value [two-sided] = 7.20 × 10^−04^) for the 128 independent endometriosis and migraine SNPs with *p*-values (*P1* and *P2*) < 0.01 (Table 1).

In an analogous analysis where the ‘IHGC migraine’ and ‘IEC endometriosis’ GWAS dataset order was reversed (designated dataset 1 and dataset 2, respectively), the number of SNP subsets with significant effect concordance remained unchanged at 144 (*P*_Fsig-permuted_ = 9.99 × 10^−04^; 95%CI: 5.12 × 10^−05^ – 5.64 × 10^−03^) and produced a similar pattern of results as before. The subset of SNPs producing the most statistically significant concordance test was *P*_assoc_ ≤ 0.6 for the ‘IHGC migraine’ and *P*_assoc_ ≤ 0.6 for the ‘IEC endometriosis’ GWAS (OR_FT_ = 1.27; *P*_FTmin-permuted_ = 1.35 × 10^−26^). 

We replicated SECA analysis using another independent migraine GWAS data from the UKBiobank. Our results confirmed significant SNP effect concordance between the ‘IEC endometriosis’ GWAS data and the ‘UKBiobank migraine’ GWAS data with 85 SNP subsets showing significant concordance of effect direction (*P*_Fsig-permuted_ = 4.0 × 10^−03^; 95%CI: 1.56 × 10 ^−03^ – 1.02 × 10 ^−02^). The most significant test was for SNP subsets with ‘IEC endometriosis’ GWAS *P*_assoc_ ≤ 0.1 and ‘UKBiobank migraine’ GWAS *P*_assoc_ ≤ 0.2 (OR_FT_ = 1.20; *P*_FTmin-permuted_ = 4.83 × 10^−04^). An analogous concordance test where the ‘UKBiobank migraine’ and ‘IEC endometriosis’ GWAS dataset order was reversed (designated dataset 1 and dataset 2, respectively) similarly indicated significant effect concordance with 119 SNP subsets producing Fisher’s exact tests with at least nominally significant concordance effects (*P*_Fsig-permuted_ = 9.99 × 10^−04^; 95%CI: 5.12 × 10^−05^ – 5.64 × 10^−03^) between the two datasets. The most statistically significant subset being SNPs with ‘UKBiobank migraine’ GWAS *P*_assoc_ ≤ 0.1 and ‘IEC endometriosis’ GWAS *P*_assoc_ ≤ 0.05 (OR_FT_ = 1.48; *P*_FTmin-permuted_ = 1.10 × 10^−04^).

### 3.2. LD Score Regression Results for Endometriosis-Migraine 

Our univariate LDSC analysis estimated SNP-based liability heritability (*h*^2^_SNP_) of 11.44% (95%CI: 10.73%–12.15%) and 8.99% (95%CI: 8.23%–9.75%), for the ‘IEC endometriosis’ and ‘IHGC migraine’ GWAS, respectively (Table 2). Cross-trait bivariate LD score regression analysis revealed a moderate, positive and highly significant genetic correlation between endometriosis and migraine (*r*_G_ = 0.38, *P* = 2.30 × 10^−25^). Using the ‘UKBiobank migraine’ GWAS data, we estimated a *h*^2^_SNP_ of 16.87% (96%CI: 15.07%–18.67%, Table 2) for migraine and a statistically significant positive correlation between the ‘IEC endometriosis’ and ‘UKBiobank migraine’ GWAS data’ (*r*_G_ = 0.14, *P* = 1.60 × 10^−3^).

### 3.3. SNPs Associated with Endometriosis and Migraine

Based on the results of our RE2 meta-analysis model (RE2 selected due to the presence of heterogeneity), 13 SNPs at one locus, associated with both endometriosis and migraine, were enriched to genome-wide signficance (*P*_SNPs_ < 5 × 10^−8^) in our meta-analysis of the ‘IEC endometriosis’ and the ‘IHGC migraine’ GWAS data (Appendix A). The 13 SNPs (rs11031005, rs11031006, rs11031040, rs11031047, rs12223987, rs12278989, rs3858429, rs4071558, rs4071559, rs4071563, rs75525300, rs7929660, rs7947350) are at a locus (the 11p14.1 locus) which has previously been reported to be genome-wide significantly associated with endometriosis (with rs74485684 as the index SNP) [31]. Indeed, all 13 SNPs are in strong LD with the endometriosis lead SNP (rs74485684), 12 having *r*^2^ > 0.8, and the remaining one, rs12223987, having *r*^2^ = 0.694. An additional 47 independent SNPs loci associated with both endometriosis and migraine showed evidence of genome-wide suggestive (*P* < 1 × 10^−5^) association (Appendix A). 

### 3.4. Mendelian Randomisation (MR)

Table 3 presents the results of the individual Wald-type ratio, IVW MR as well as the various sensitivity analyses—summarising the association of our IVs with endometriosis and migraine. Given the large sample size of our ‘IEC endometriosis’ GWAS data, the robust association between our IVs and endometriosis and the approximate F-statistics greater than 30, our IVs are strong and are not expected to suffer from weak instrument bias [58]. 

Nevertheless, we found evidence for marginally significant heterogeneity among the IVs (Cochran’s Q statistics for IVW = 18.39, degree of freedom [df] = 10, *P* = 0.049, and Q’ statistics for MR Egger = 16.99, df = 9, *P* = 0.049). Under the null hypothesis of no heterogeneity, we expect that the value of Q and Q′ will be same as their corresponding df (10 and 9, respectively). This is not the case. However, the difference between Q and its df (18.39 – 10 = 8.39) for IVW and (Q’, 16.99 – 9 = 7.99) for MR Egger are small, indicating (alongside the borderline significant *p*-value) that the heterogeneity was not substantial. In addition, we did not find MR Egger a better fit for our data than the IVW model since the difference *Q* − *Q*^′^ = 1.4 is not sufficiently extreme under a $\chi_{1}^{2}$ distribution. Our selected instruments reportedly explained about 1.75% variance in endometriosis [31].

Combining all the 11 endometriosis SNPs (IVs), MR analysis did not find evidence for a causal relationship between endometriosis and migraine based on the IVW method ([OR = 0.98, 95%CI: 0.89 – 1.07, *P* = 0.667] per standard deviation increase in endometriosis risk). Our results for sensitivity analyses using the MR-Egger (OR = 0.78, 95%CI: 0.46 – 1. 32, *P* = 0.381), and the weighted median (OR = 0.92, 95%CI: 0.83 – 1.02, *P* = 0.098) agree with that of the IVW method. Furthermore, the MR-Egger intercept (representing the average estimate of the pleiotropic effects of a SNP) was 0.0232 (SE: 0.027, *P*: 0.413). This intercept was not significantly different from zero, indicating that there was no evidence of directional (unbalanced horizontal) pleiotropy. However, our results for ‘single SNPs MR’ analysis identify *rs74485684* to be statistically significant as endometriosis genetic variant with risk-increasing effect on migraine (OR= 1.43, 95%CI: 1.11 – 1.83, *P* = 0.006). This SNP was nominally associated with migraine (Table 3) as well as, ‘length of menstrual cycle’ and ‘excessive, frequent and irregular menstruation’ (Appendix A). The results of MR excluding the SNP (data not shown) did not make any difference to our previous finding, supporting the evidence of no causal association between our exposure and outcome variables.

Compared to endometriosis, a greater number of genome-wide significant SNPs have been identified for migraine [31,32]. Consequently, we conducted a “TwoSampleMR” utilising independent genome-wide significant SNPs from migraine GWAS as IVs, migraine as the exposure variable, and endometriosis as the outcome variable, reversing the direction of the datasets (data not shown). We note, however, that the causal effects of migraine on endometriosis may be difficult to explain conceptually. Regardless, the results for this analysis also did not provide evidence for a causal association between migraine and endometriosis.

### 3.5. Gene-Based Analysis for Endometriosis and Migraine

Our gene-based association analyses identified 1,749 and 1,871 genes nominally significant (*P*_gene_ < 0.05) in the ‘IEC endometriosis’ and ‘IHGC migraine’ GWAS gene-level association results, respectively (Appendix A). A Bonferroni adjustment using the largest estimated total effective number of genes (17,104) produced a genome-wide, gene-based threshold of 2.92 × 10^−6^ (0.05/17,104). At this threshold, nine genes (*ARL14EP, VEZT, CDC42, LINC00339, WNT4, GREB1, IL1A, FGD6, KDR*) were genome-wide significant (Appendix A) in the gene-based analysis for the ‘IEC endometriosis’ GWAS, all of which have previously been reported for endometriosis (assessed using PhenoScanner v2 [49] (http://www.phenoscanner.medschl.cam.ac.uk, on 30 September 2019). Similarly, for migraine, a total of 17 genes (*PLCE1, PLCE1-AS1, MRVI1, LRP1, STAT6, MEF2D, PRDM16, MROH2A, TRPM8, POC5, FHL5, KCNK5, PHACTR1, UFL1, TMEM91, MSL3P1, ANKDD1B*) were genome-wide significant (Appendix A) in the gene-based analysis (at 2.92 × 10^−6^ threshold). Following an assessment in PhenoScanner v2 (accessed on 22^nd^ September 2019), five of the 17 genes (*PLCE1-AS1, MROH2A, POC5, TMEM91, ANKDD1B*) have not previously been reported for migraine. Of the five new migraine genes, two (MROH2A on chromosome 2q37.1, and PLCE1-AS1 on chromosome 10q23.33) were located at previously reported migraine loci. The remaining three genes are located at two loci not previously identified for migraine (POC5 and ANKDD1B on chromosome 5q13.3, and TMEM91 on chromosome 19q13.2), thus, representing novel loci for migraine risks.

We assessed overlapping genes between endometriosis and migraine using gene-based test outputs, and our results revealed a total of 17 (at *P*_gene_ < 0.01), 196 (at *P*_gene_ < 0.05), and 493 (at *P*_gene_ < 0.1) significantly enriched genes shared by the two traits (Appendix A, respectively). Moreover, following FCP estimation for overlapping genes at *P*_gene_ < 0.1, three genes, *ARL14EP* (on chromosome 11p14.1), *TRIM32* (on chromosome 9q33.1), and *SLC35G6* (on chromosome 17p13.1) were enriched to a genome-wide significant level based on their combined *p*-value (Table 4). Two of these three genes (*TRIM32*, and *SLC35G6*) were not genome-wide significant in endometriosis or migraine; rather they attained genome-wide significance following the combination of the respective gene association *p*-values for the two traits indicating evidence of their involvement in the two disorders (and possibly their comorbid state). *ARL14EP*, on the other hand, was genome-wide significant for endometriosis only but attained a more genome-wide significance status following the estimation of FCP using both endometriosis and migraine gene *p*-values. 

Lastly, the exact binomial test confirms that significant gene-based genetic overlap exists between endometriosis and migraine at *p*-value thresholds of *P* < 0.1 and *P* < 0.05 (Table 5). For example, at gene-based *p*-value < 0.05, the observed proportion of genes overlapping the two traits (12%) was significantly higher than the expected proportion (8.6%) [*P*_binomial-test_ = 9.83 ×10^−06^]. These results indicate that the observed gene-based genetic overlap between endometriosis and migraine was more than expected by chance implying that at the least, a proportion of the identified overlapping genes are truly associated with both endometriosis and migraine. 

### 3.6. Functional Enrichment Analyses

Functional enrichment analysis identifies six significantly enriched biological pathways for the 493 genes overlapping endometriosis and migraine at *P*_gene_ < 0.1. Table 6 presents a summary of these pathways. Clusters were generated following enrichment mapping and auto-annotation thereby collapsing the identified pathways into three main biological themes and clusters: mitogen-activated protein kinase (MAPK) signalling pathway, regulation of kappa-light-chain-enhancer of activated B cells (**kappaB**) signalling and tumor necrosis factor (TNF) alpha signalling pathway (Figure 1). 

## 4. Discussion

Several observational epidemiological studies have reported the comorbidity of endometriosis with migraine. For the first time, however, we present a comprehensive assessment of the molecular genetic overlap, causal relationship as well as shared genes and biological pathways between the two disorders. SECA reveals the existence of a strong and significant genetic overlap between endometriosis and migraine. For instance, the proportion of nominally significant (*P* < 0.05) independent SNPs with concordant risk allele effects for endometriosis and migraine (58%) was higher than expected under the null hypothesis of no association (*P*_concordant_ = 2.23 × 10^−9^). Bivariate LDSC analysis estimates a moderate, positive and highly significant genetic correlation between endometriosis and migraine (*r*_G_ = 0.38, *P* = 2.30 × 10^−25^). Notably, we reproduced these significant findings using a second independent migraine GWAS dataset from the UKBB (*r*_G_ = 0.14, *P* = 1.60 × 10^−3^). The weaker genetic correlation observed in the latter is most likely due to the smaller sample size (migraine cases) and the broader ‘self-reported migraine’ phenotype in the ‘UKBB migraine’ GWAS.

Our finding of significant genetic overlap and correlation between endometriosis and migraine indicates the presence of shared genetic components between the two disorders and confirms their comorbidity. This means that endometriosis patients share a non-negligible proportion of genetic risk variants with migraine patients. The SNP-based heritability estimated for endometriosis and migraine were lower than those reported from the twin-based studies due to the imperfect tagging of causal variants by common SNPs, in particular, if the causal variants are rare [59]. However, our findings compare favourably with those of a previous twin-based study which concluded that common genetic influences explain the comorbidity of migraine and endometriosis [13]. 

Although a meta-analysis of migraine and endometriosis GWAS produced a number of SNPs with genome-wide significant *p*-values, no novel risk loci were identified as all 13 SNPs reported were in strong LD with a previously reported risk locus for endometriosis on 11p14.1. Our finding, nonetheless, indicates the potential involvement of the locus in both disorders, and possibly, in their comorbid state. In addition to endometriosis, the 11p14.1 locus comprising *FSHB* has been associated with several female hormone-related traits including age at menarche and menopause, short menstrual cycle, polycystic ovarian syndrome, and increased risk of dizygotic twinning [31,60,61,62]. Thus, the locus may influence risk for both endometriosis and migraine via more frequent (menstrual-related) hormonal fluctuations in women as the same variants at this locus are associated with shorter and more frequent menstrual cycles and influencing oestradiol release, which have both been implicated in migraine risk [31,60]. We identified an additional 47 independent SNPs loci enriched to genome-wide suggestive (*P* < 1 × 10^−5^) association which should be prioritised in future studies. Meta-analysing more powerful GWAS data (with larger sample sizes) for endometriosis and migraine will identify more robust SNPs and novel risk loci shared by the two disorders. 

Results for ‘single SNPs MR’ analysis showed evidence that one of the 11 endometriosis genome-wide significant SNPs, rs74485684, had a statistically significant risk-increasing effect on migraine. The rs74485684 SNP is located on chromosome 11p14.1 near *FSHB* gene, a locus significantly enriched for both endometriosis and migraine in our meta-analysis. The results for our PhenoScanner analysis, however, indicate a significant and strong association between rs74485684 SNP and some traits namely ‘length of menstrual cycle’ and ‘excessive, frequent and irregular menstruation’. There is evidence that the named traits represent important risk factors for both endometriosis and migraine [13,14,15,16,17,18,19] which may have confounded our MR analysis—i.e., a violation of the second assumption of MR analysis [48]. This observation would negate a causal relationship of the endometriosis SNP (rs74485684) on migraine but lend support for a ‘shared genetic risk factor mechanism of association’ in the comorbidity of the two disorders. 

Combining all 11 endometriosis risk SNPs, MR analysis did not provide evidence of a causal relationship between endometriosis and migraine. We note, however, that the variance in endometriosis explained by the combined multi-allelic instrument is rather small (less than 2%) indicating that our MR estimates were biased towards the null [63]. Thus, we cannot completely rule out the possibility of causal effects of endometriosis on migraine. Future studies should revisit the MR analysis when more genome-wide significant SNPs associated with endometriosis are available. Although we do not have evidence of a causal relationship between endometriosis and migraine, some other mechanisms of association may explain their co-occurrence. For example, observational studies have identified some epidemiological similarities for both endometriosis and migraine [13,14,15,16,17,18,19], suggesting a ‘shared risk factor mechanism of association’. The results of our genetic overlap analyses support this position—identifying shared genetic risk factors for the co-occurrence of endometriosis and migraine. 

Moving beyond the SNP-level study, we conducted gene-based analyses thereby furthering our assessment of the genetic overlap between endometriosis and migraine. Considered the basic physical and functional unit of the human genome, genes exhibit a closer relationship with biological mechanisms than SNPs. Moreover, gene-based analyses have the ability to account for LD and allelic heterogeneity while examining the association between a trait of interest and multiple co-located SNPs [64]. Thus, gene-based methods can provide a more robust and interpretable approach to understanding the biology of complex traits [64]. Like the SNP-based analysis, we found a significant gene-level genetic overlap between endometriosis and migraine with a total of 196 significantly enriched genes nominally associated (*P*_gene_ < 0.05) with both traits (*P*_binomial-test_ = 9.83 × 10^−6^). Three overlapping genes, *ARL14EP* (on chromosome 11p14.1), *TRIM32* (on chromosome 9q33.1), and *SLC35G6* (on chromosome 17p13.1), were genome-wide significant based on their combined gene association *p*-values. We note, nonetheless, that these results are based on a statistical association of variants in and directly flanking each gene and do not strictly functionally implicate the genes. The 11p14.1 locus harbouring the *ARL14EP* gene has previously been associated with endometriosis [31], and implicated in our cross-disorder meta-analysis as well as the ‘single SNPs MR’ analysis (present study). However, the roles of the gene in migraine as well as in the comorbidity of endometriosis and migraine remain to be elucidated. *ARL14EP* is well expressed in thyroid and adrenal glands, brain, endometrium, lymph nodes, ovary, and many other tissues. More targeted studies are now warranted for a clearer understanding of the gene and its relationship with both endometriosis and migraine. 

The remaining two genome-wide significant genes, *TRIM32 and SLC35G6,* have not been previously reported for endometriosis or migraine, neither are they located at or near established loci for any of the two disorders; hence, they represent two novel genes and susceptibility loci for the two traits. The *SLC35G6* gene is well expressed in the testis, lowly expressed in the endometrium and adrenal gland; however, information about its biological functions is limited; hence, further investigation into the gene and its involvement in endometriosis and migraine is necessitated. Conversely, *TRIM32* is a protein-coding gene consisting of a ‘RING, B-box, coiled-coil and six C-terminal NHL domains’ [65]. Being a ubiquitously expressed E3 ligase, the gene targets several proteins for degradation through ubiquitination [65]. *TRIM32* has broad substrate specificity and has been associated with several biological activities including the regulation of microRNA, tumorigenesis, development and differentiation, as well as innate immunity. Moreover, *TRIM32* has been linked with certain disorders such as Bardet–Biedl syndrome (mutation in ‘the B-box’ domain of the gene) [66,67], and limb-girdle muscular dystrophy (mutations in the ’C-terminal NHL domain’ of the gene) [68]. Endometrium, adrenal gland and the brain are among the three leading sites of *TRIM32’s* expression, lending greater support for its potential involvement in endometriosis and migraine. More targeted studies are required to elucidate *TRIM32’s* exact role in the two traits. 

To explain the pathogenesis of co-occurring endometriosis and migraine, some authors have suggested a number of possible biological mechanisms including the roles of elevated levels of circulating prostaglandins [27], hormonal fluctuations [13], and impaired regulation of nitric oxide synthesis [69,70]. The involvement of hormone fluctuations is especially favoured by the fact that both endometriosis and migraine share risk factors consistent with the hormone-based regulation of the menstrual cycle such as early menarche, and menorrhagia [17,18,19]. The results of our meta-analysis, ‘single SNPs MR’ and, partly, overlapping genes assessment, potentially support a role for sex hormones activities in the pathogenesis of the two disorders. Following functional enrichment analysis, we also found significantly enriched biological pathways shared by both traits that may differ in some respects from the aforementioned mechanisms. For ease of interpretation as well as to eliminate possible redundancy, we carried out enrichment mapping thereby collapsing the over-represented pathways into three simplified biological themes and clusters. The first cluster of biological pathways, MAPK signalling pathway, comprises ‘focal adhesion-PI3K-Akt-mTOR-signaling’ and ‘MAPK signalling’. MAPK and ‘PI3K-Akt-mTOR’ are expressed differently, however, there is evidence that both are activated by steroid hormones and growth factors [71], supporting a role for sex hormones in the pathogenesis of endometriosis and migraine. 

‘Focal adhesion-PI3K-Akt-mTOR’ is a signalling cascade made up of ‘focal adhesion’ (or cell-matrix adhesions), ‘phosphatidylinositide 3 kinases’ (PI3K), ‘protein kinase B’ (AKT), and ‘mammalian target of rapamycin’ (mTOR). Besides their structural role of mediating the molecular contact between intra- and extra-cellular spaces [63], focal adhesions relay signals between cells and the extracellular matrix, consequence upon which a range of cellular responses—cell growth, differentiation and movement—are initiated [63]. Protein kinases and phosphatases—two opposing but complementary groups of cells signalling proteins—as well as integrins, constitute essential parts of focal adhesive molecules [72,73]. While kinases and phosphatases co-regulate protein phosphorylation, a biological process that is critical to several cellular functions, integrins sense the environment and subsequently evoke responses resulting in the regulation of cell motility and shapes [72,73]. Also, through a complex interplay of its core components—PI3K stimulation, AKT phosphorylation, mTOR activation—the ‘PI3K-Akt-mTOR’ pathway facilitates several cellular processes including cell proliferation, metabolism, angiogenesis, and apoptosis [74]. There is evidence implicating these mechanisms in the biology of endometriosis and migraine [71,75]. 

For example, the role of kinases, particularly, MAPK, is well supported in the causal pathway of endometriosis, and arguably migraine [64,65,66,67]. Altered peritoneal microenvironment caused by endometriotic lesions is believed to activate kinase signalling pathways which may result in kinase-dependent growth or proliferation of endometriotic lesions [68]. In the case of migraine, activation of MAPK is suggested to mediate the synthesis and the release of calcitonin gene-related peptide (CGRP) which has long been implicated in the pathophysiology of migraine [66,69]. Indeed, recently approved monoclonal antibodies (mAbs) targeting CGRP or its receptor have lately been developed, representing a first major breakthrough for migraine-specific treatments in 30 years [76,77,78]. Furthermore, overexpression of ‘PI3K-Akt-mTOR’ has been noted in endometriosis and certain types of cancers (ovarian, breast and urothelial), and therapeutic agents targeting its core components have been developed [71,79]. 

‘Interleukin-1 receptor binding’ and ‘regulation of I-kappaB kinase and NF-kappaB signalling’, converged to a second biological cluster of pathways, regulation of kappaB signaling, following enrichment mapping and auto-annotation. Nuclear factor- kappaB (‘NF-kappaB’) is a transcription factor regulating inflammatory responses and mediating several functions of both the adaptive and innate immunity [80,81]. In addition to participating in the regulation of inflammatory processes, ‘NF-kappaB’ plays an important role in the expression of certain pro-inflammatory genes such as those involved in coding for cytokines [81]. Interleukin 1, on the other hand, is a pro-inflammatory cytokine whose activities are mediated through interleukin-1 receptor binding [82,83]. There are two types of this receptor: interleukin-1 receptor I which mainly transmits inflammatory signals, and interleukin-1 receptor II which although transmits no signals may suppress the effects of interleukin-1 by competing for its active binding sites [82,84]. Interleukin 1 not only mediates innate immune reactions, it also activates ‘NF-kappaB’ inflammatory pathways [83,84]. Thus, in line with previous studies [83,85], our study suggests that inflammatory processes and immune system dysfunction, mediated by the deregulation of cytokines and the ‘NF-kappaB’ factor [81], maybe relevant in the causal pathways of endometriosis and migraine. 

Lastly, the tumour necrosis factor-alpha (TNF-α) signalling pathway was significantly enriched as one of the biological mechanisms underpinning endometriosis and migraine in the present study. Primarily produced by activated macrophages, T helper type 1 cells and natural killer cells, TNF-α, is among the most studied member of the TNF family [86,87]. The protein acts commonly alongside interleukin-1 and similarly activates the ‘NF-kappaB’ inflammatory pathways [86,87]. Consistent evidence indicates that women with endometriosis have a higher level of TNF-α in their peritoneal fluid and endometrium [83,88]. Also, the size of endometriotic lesions has been reported to be positively correlated with the concentration of TNF-α [83,88]. Therefore, our finding agrees with previous studies which have recognised the role of TNF in the pathogenesis of endometriosis [83,88]. In contrast, contradictory evidence for the role of TNF-α in migraine has been reported [89,90]. Hence, the present study provides important support for TNF-α in both endometriosis and migraine pathogenesis. 

### Strengths and Limitations

Major strengths of this study include our use of multiple statistical methods in analysing well-powered world-leading datasets to provide a comprehensive assessment of the relationship between endometriosis and migraine at the molecular genetic level. Furthermore, being based on genotype data, these analyses are generally not susceptible to potential confounding effects often associated with observational studies, thus providing strong and reliable evidence in support of our findings. For example, unlike in the traditional observational studies where the confounding effects of lifestyles or environmental factors are highly likely, genotypes are known to be well established and fixed at conception and should not be confounded by lifestyles or environments. Also, given that the inheritance of genotype precedes exposure to environmental factors, and, hence, disease onset later on in the offspring, the possibility of reverse causality is avoided in our study, lending credence to our findings. Limitations of our study mainly relate to those specific to the analysis methods. For example, sample overlap may confound LDSC and MR analyses. However, we ensured the independence of our samples and used a range of recommended approaches to minimise a possible violation of the MR assumptions. Lastly, several of the significantly enriched mechanisms in the pathway-based analyses are prone to redundancy. To minimise this limitation, however, we performed enrichment mapping and auto-annotation to collapse related pathways to simplified biological themes and clusters, thereby, enhancing the visualisation and interpretation of the significantly enriched biological pathways. 

## 5. Conclusions

Our findings further confirm the comorbidity of endometriosis and migraine and indicate a non-causal relationship between the two traits, with shared genetically controlled biological mechanisms underlying the co-occurrence of the two disorders. After combining gene-based *p*-values across endometriosis and migraine GWAS, we found that three genes (*ARL14EP*, *TRIM32*, and *SLC35G6*), were genome-wide significant. Two of these genes (*TRIM32*, and *SLC35G6)* have not previously been reported for endometriosis or migraine, nor were they located on or near previously identified loci for any of the two traits—indicating that they represent novel genes and susceptibility loci for both endometriosis and migraine. Our functional enrichment analyses reveal some genetically controlled biological pathways underlying endometriosis and migraine including interleukin-1 receptor binding, focal adhesion-PI3K-Akt-mTOR-signaling, MAPK and TNF alpha signalling. Biological mechanisms related to sex hormone activities, protein adhesion and phosphorylation as well as inflammatory and immune system dysfunction, among others, are implicated by these pathways. Our study further supports the importance of a concurrent screening for migraine in patients presenting with or being investigated for endometriosis. Clinicians, thus, would need to start exercising a heightened suspicion for migraine in endometriosis patients. Shared genes and biological pathways identified in the present study could serve as potential therapeutic targets for endometriosis and migraine and perhaps the comorbid state of the two traits. However, further molecular and functional studies are needed for a targeted investigation into their roles in both disorders. Future analyses utilising results from more powerful GWAS are expected to improve the power to identify more robust SNPs and loci, as well as genes for endometriosis, migraine and the co-occurrence of the two disorders.

## Figures and Tables

**Figure 1 genes-11-00268-f001:**
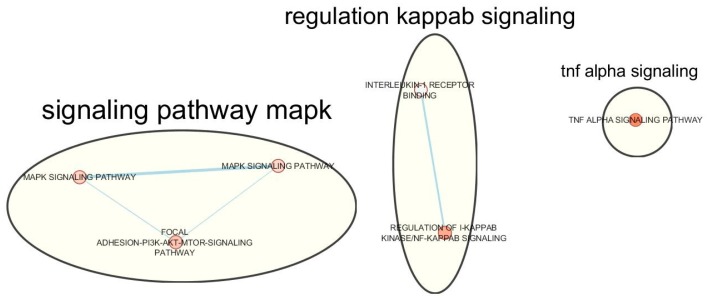
Clustered biological themes for overlapping endometriosis–migraine genes.

**Table 1 genes-11-00268-t001:** SNP effect concordance analysis (SECA) results for the test of genetic concordance between endometriosis and migraine.

*P1*	*P2*	Concordant SNPs	Discordant SNPs	Total SNPs	Proportion of Concordance	OR	*P*
**1**	1	30790	28398	59188	0.52	1.18	8.77 × 10^−23^
**0.9**	0.9	27540	25213	52753	0.52	1.19	4.11 × 10^−24^
**0.8**	0.8	24222	21971	46193	0.52	1.22	1.30 × 10^−25^
**0.7**	0.7	20842	18665	39507	0.53	1.25	6.60 × 10^−28^
**0.6**	0.6	17474	15458	32932	0.53	1.28	1.23 × 10^−28^
**0.5**	0.5	14218	12383	26601	0.53	1.32	2.69 × 10^−29^
**0.4**	0.4	10973	9415	20388	0.54	1.36	1.31 × 10^−27^
**0.3**	0.3	7804	6510	14314	0.55	1.44	3.21 × 10^−27^
**0.2**	0.2	4771	3855	8626	0.55	1.53	6.92 × 10^−23^
**0.1**	0.1	1946	1496	3442	0.57	1.69	2.06 × 10^−14^
**0.05**	0.05	804	579	1383	0.58	1.92	2.23 × 10^−09^
**0.01**	0.01	85	43	128	0.66	3.61	7.20 × 10^−04^

*P1*: International Endogene Consortium (IEC) Endometriosis data *p*-value; *P2*: International Headache Genetics Consortium (IHGC) migraine data *p*-value; SNP: single nucleotide polymorphism; OR: odds ratio for the effect direction concordance association test for endometriosis and migraine; *P*: Fisher’s exact *p*-value for the effect direction concordance association test between endometriosis and migraine.

**Table 2 genes-11-00268-t002:** Linkage disequilibrium (LD) score regression summary.

**SNP-based Heritability**						
**Phentype**		**Dataset source**	**Valid SNPs in analysis**	**Liability scale *h*^2^_SNP_ (95% CI)**		**Intercept (se)**
**Endometriosis**		IEC	1,157,235	11.44% (10.73–12.15%)		Constrained to 1
**Migraine**		IHGC	1,173,223	8.99% (8.23–9.75%)		1.0232 (0.008)
**Migraine**		UKBB	1,177,705	16.87% (15.07–18.67%)		1.0122 (0.007)
**SNP-based Genetic Correlation**						
**Phenotype 1** **(data source)**	**Phenotype 2** **(data source)**	**SNPs with valid alleles**	***r*_G_ (se)** **[*p*-value]**	**Phenotype 1** **Intercept**	**Phenotype 2** **Intercept**	**Gencov** **Intercept (se)**
**Endometriosis (IEC)**	Migraine (IHGC)	1,154,255	0.38 (0.0364) [2.30 × 10^−25^]	Constrained to 1	1.0214 (specified)	Constrained to 0
**Endometriosis (IEC)**	Migraine (UKBB)	1,152,558	0.14 (0.0438) 1.60 × 10^−^°^3^	Constrained to 1	1.0136 (specified)	Constrained to 0

IEC: International Endogene Consortium, IHGC: International Headache Genetics Consortium, UKBB: United Kingdom BioBank, SNP: single nucleotide polymorphism, *h*^2^_SNP_: SNP-based heritability, CI: confidence interval, se: standard error.

**Table 3 genes-11-00268-t003:** Instrumental variables, Mendelian randomisation (MR) results and sensitivity analyses.

**SNPs**	**EA**	**OA**	**Beta (endo)**	**SE (endo)**	**P(endo)**	**Beta (migr)**	**SE (migr)**	**P(migr)**	**Beta (endo-migr)**	**F-Stat ***	**SE (endo-migr)**	**P (endo-migr)**
rs10167914	A	G	−0.11	0.02	1.10 × 10^−^°^9^	0.01	0.01	0.27	−0.11	37.27	0.10	0.27
rs11674184	T	G	0.12	0.01	2.67 × 10^−17^	−0.02	0.01	0.16	−0.13	71.40	0.09	0.16
rs12037376	A	G	0.15	0.02	8.87 × 10^−17^	0.01	0.01	0.35	−0.09	68.97	0.10	0.35
rs12700667	A	G	0.10	0.02	9.08 × 10^−1^°	0.01	0.01	0.6	−0.07	37.64	0.13	0.61
rs1537377	T	C	−0.09	0.01	1.33 × 10^−1^°	0.00	0.01	0.83	0.03	41.53	0.12	0.83
rs1903068	A	G	0.10	0.01	1.04 × 10^−11^	0.01	0.01	0.32	0.11	46.28	0.11	0.32
rs4762326	T	C	0.08	0.01	2.20 × 10^−^°^9^	−0.01	0.01	0.31	0.14	35.55	0.13	0.31
rs6546324	A	C	0.08	0.01	3.01 × 10^−^°^8^	−0.01	0.01	0.42	0.11	30.56	0.14	0.42
rs71575922	C	G	−0.11	0.02	2.02 × 10^−^°^8^	−0.01	0.01	0.35	−0.12	31.41	0.13	0.35
rs74485684	T	C	0.11	0.02	2.00 × 10^−^°^8^	0.04	0.01	0.01	0.36	31.55	0.13	0.01
rs760794	T	C	0.09	0.01	1.79 × 10^−1^°	−0.02	0.01	0.07	−0.22	40.43	0.12	0.07
**Methods**			**Number of SNPs**		**Beta**		**SE**			**P**		
All—IVW			11		−0.02		0.05			0.67		
All—MR Egger			11		−0.25		0.27			0.38		
All—Simple mode			11		0.08		0.12			0.50		
All—Weighted mode			11		−0.11		0.07			0.14		
All—Weighted median			11		−0.09		0.05			0.10		

SNP: single nucleotide polymorphism, endo: endometriosis, migr: migraine, EA: effect allele, OA: other allele, Endo-migr: endometriosis as exposure and migraine as the outcome variable, Beta: effect size in standard deviation unit, SE: standard error, P: *p*-value, IVW: inverse variance weighted; * we estimated approximate F statistics values using t-statistics = Beta/SE, which is the t distribution with N-1 degrees of freedom (N is our sample size). The square of the t statistic represents approximate F statistics with degrees of freedom = 1. Thus, approximate F-statistics = (Beta_endo/SE_endo)^2^.

**Table 4 genes-11-00268-t004:** Genome-wide significant genes overlapping endometriosis and migraine in gene-based association analyses.

S/N	Chro	Gene	Start Position	Stop Position	IEC Endometriosis			IHGC Migraine			
					Gene *p*-Value	Top SNP	Top SNP *p*-Value	Gene *p*-Value	Top SNP	Top SNP *p*-Value	FCP
1	11	ARL14EP	30344648	30359165	1.00 × 10^−^^0^^6^	rs4071559	5.60 × 10^−08^	5.54 × 10^−02^	rs4071559	5.97 × 10^−03^	9.81 × 10^−07^
2	9	TRIM32	119449580	119463579	2.76 × 10^−02^	rs11793648	3.14 × 10^−03^	5.00 × 10^−06^	rs76973802	7.15 × 10^−07^	2.32 × 10^−06^
3	17	SLC35G6	7384720	7386383	1.59 × 10^−03^	rs9891297	3.09 × 10^−04^	9.40 × 10^−05^	rs8065577	2.21 × 10^−05^	2.50 × 10^−06^

Chr: chromosomes; FCP: Fisher’s combined *p*-value; IEC: International Endogene Consortium; IHGC: International Headache Genetic Consortium.

**Table 5 genes-11-00268-t005:** Summary of gene-level association analyses for endometriosis and depression under three *p*-value thresholds.

**The effective number of genes in Endometriosis and migraine**											
**Disorder**	**Total genes**		***P* value < 0.1**			***P* < 0.05**			***P* < 0.01**		
**Raw ^c^**	**Effective ^d^**	**Raw ^c^**	**Effective ^d^**	**Proportion ^e^**	**Raw ^c^**	**Effective ^d^**	**Proportion ^e^**	**Raw ^c^**	**Effective ^d^**	**Proportion ^e^**
Endometriosis ^a^	20473	17104	2966	2433	0.142	1749	1430	0.084	481	386	0.023
Migraine ^b^	20473	17046	3239	2579	0.151	1871	1467	0.086	587	450	0.026
**Number of overlapping genes and binomial test results for gene-based association**											
**Discovery**	**Targets**		**Overlapping genes**		**Proportion of overlap**				**Binomial test *P*-value**		
			**Raw**	**Effective**	**Expected**		**Observed**				
***P* value < 0.01**						
Endometriosis	Migraine		17	15	450/17046 = 0.026		15/386 = 0.039		0.08259		
***P* value < 0.05**						
Endometriosis	Migraine		196	171	1467/17046 = 0.086		171/1430 = 0.120		9.83 × 10^−06^		
***P* value < 0.1**						
Endometriosis	Migraine		493	420	2579/17046 = 0.151		420/2433 = 0.173		1.85 × 10^−03^		

^a^ Endometriosis data from International Endogene Consortium, ^b^ migraine data from International Headache Genetic Consortium (IHGC), ^c^ raw number of genes, ^d^ effective number of independent genes, ^e^ proportion of total effective number of genes.

**Table 6 genes-11-00268-t006:** Significantly enriched ordered pathways for overlapping endometriosis-migraine genes.

**Term ID for pathway**	**Pathway term name**	**Adjusted *p*-value**	**Genes**
**Source: Gene Ontology (Molecular function)**			
Interleukin-1 receptor binding	GO: 0005149	9.19 × 10^−03^	*IL36RN, IL37, IL36B, IL1B, IL1F10*
**Source: Gene Ontology (Biological process)**			
Regulation of I-kappaB kinase and NF-kappaB signalling	GO: 0043122	1.90 × 10^−02^	*TRIM32, IL36RN, IL37, TMED4, IL36B, IL1B, RNF31, IKBKB, SHISA5, TANK, PARK2, IL1F10, ZDHHC17, GSTP1, DAB2IP, SLC35B2, TRIM13*
**Source: Biological pathways (Kyoto Encyclopedia of Genes [KEGG])**			
MAPK signalling pathway	KEGG: 04010	1.40 × 10^−02^	*IL1B, FGF18, NGF, IKBKB, MAP2K5, PTPN5, PDGFC, MAPK9, NRAS, PPP3CA, CACNA1E, FGF17, MAP2K6, FGF9, MET, RPS6KA4, FGFR4*
**Source: Biological pathways (WikiPathways)**			
MAPK Signalling Pathway	WP: WP382	1.3 × 10^−02^	*FGF11, IL1B, FGF18, NGF, IKBKB, MAP2K5, PTPN5, MAPK9, NRAS, PPP3CA, CACNA1E, FGF17, MAP2K6, FGF9, RPS6KA4, FGFR4*
Focal Adhesion-PI3K-Akt-mTOR-signaling pathway	WP: WP3932	1.54 × 10^−02^	*FGF11, ITGB5, CREB5, PFKFB4, PPP2CA, DDIT4, FGF18, NGF, IKBKB, PTK2, PDGFC, SLC2A4, NRAS, CREB3L2, FGF17, FGF9, MET, FGFR4*
TNF alpha Signalling Pathway	WP: WP231	2.3 × 10^−02^	*PPP2CA, RFK, IKBKB, PTPRCAP, MAPK9, NSMAF, NRAS, TANK, MAP2K6*

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
