# Peer review of "Shared Molecular Genetic Mechanisms Underlie Endometriosis and Migraine Comorbidity"

_genes, 2020, doi:10.3390/genes11030268_

Round 1

Reviewer 1 Report

Overall, an interesting and well-performed study assessing the genetic overlap between endometriosis and migraine, with believable results and proper discussion of caveats and limitations.

1-Needs to be carefully proofread to fix grammar mistakes and typos

2-There are many places where slashes are used (/), which seems inappropriate in scientific text

3-What is the advantage of performing SECA over just using LDSC?

4-In section 3.3. give the number of LOCI that were associated significantly, not the number of SNPs. Are all 13 SNPs at a single locus or across several? This is an important detail.

5-What is the LD for your signal on 11p14.1 with the previously reported endometriosis locus?

6-Again, for the suggestive SNPs, it makes more sense to resolve these into discrete loci and report that number rather than just the number of SNPs, which is highly dependent on reference map used. You could do the same analysis with HapMap and 1000G reference panels, and have the same number of risk LOCI appear, but the HapMap set would have far fewer SNPs because the reference panel contains fewer SNPs. For example both analyses might result in 5 risk LOCI, but the HapMap analysis would have 10 significant SNPs and the 1000G would have 100.

7-MR has been performed thoroughly and conclusions are appropriate

8-This is phrased oddly to me: “This means that a non-negligible proportion of endometriosis patients share genetic susceptibility with migraine patients”—don’t you mean that a non-negligible proportion of endometriosis RISK shares genetic susceptibility with migraine RISK?

9-“ However, the SNP-based heritability estimated for endometriosis and migraine in the present study were relatively lower than those reported in the twin-based study. This discrepancy may be explained by the inability of the present GWAS sample sizes to capture all the risk SNPs, some of which may be rare or have small effects, for endometriosis and migraine, a phenomenon referred to as the ‘missing heritability’”

I have several issues with these statements. First, the discrepancy between twin-based heritability and SNP-based heritability is that not every single possible variant is included in SNP-based heritability: only included are the SNPs you put in (i.e. directly genotyped and imputed SNPs). Thus, family-based heritability captures more variance than SNP-based heritability because there is no filter based on SNP coverage. This is not really dependent on GWAS sample size but rather SNP coverage and frequency cutoffs. Yes, this does mean that rarer variants of higher effects are likely omitted from SNP-based heritability. Second, I do not think it is necessary to go into “missing heritability”. These are just different estimates based on different input parameters. It is well understood and accepted that SNP-based heritability estimates are almost always lower than family-based estimates.

Reviewer 2 Report

Comments:

In this original manuscript, Adewuyi et al. demonstrated for the first time that endometriosis and migraine have shared genetic components with overlapping genes corresponding to specific biological signaling pathways. Even though genetic analyses in the manuscript did not find any genome-wide novel SNPs and causal relationships associated with endometriosis and migraine, the authors found two novel loci in gene-level association studies between endometriosis and migraine. Additionally, functional enrichment analyses exhibited possible biological pathways significantly enriched in both traits. This research is significant in the context of linking endometriosis and migraine together via similar molecular pathways.

The introduction of the manuscript is nicely written. But it can be improved as suggested below. Methods were explained very diligently along with excellent experimental design. The results were also written very precisely. The discussion section, however, needs some major additions as suggested below. Grammatical errors were minor and mentioned in my suggestions.

More importantly, along with the suggestions below, I am more concerned regarding the medication history and other medical conditions of the patients recruited in the datasets utilized in the manuscript. Oral contraceptives, hormone pills, NSAIDS, etc. can drastically increase comorbidity of endometriosis and migraine. Diseases like cancer, depression, etc. can further increase the medications’ list and impact gene expression. I request authors to kindly provide medications taken and medical history of patients while recruited, if accessed and available, or else if any adjustment has been made computationally for the same or mention it as a weakness of the study if data can not be available.

Additionally, enrichment analyses showed few molecular pathways upregulated in endometriosis and migraine patients. Can the authors show common pathways among both the traits that are downregulated significantly? If yes, then it will add significant value to the manuscript. If not, then kindly provide a necessary explanation.

Overall, this manuscript provides novelgene loci and common underlying biological pathways, further confirming the already established, co-occurrence of endometriosis and migraine.

Suggestions:

  1. Line 31- examined
  2. Line 37- no comma needed
  3. Line 38- “the” gene-based analysis
  4. Line 44- "Our findings further confirm..."
  5. Lines 52-64- In the first paragraph of the introduction, it would be helpful if authors can provide the most up-to-date statistics of endometriosis and migraine. The references indicate the statistics are at least 5-10 years old.
  6. Lines 45 and 546- contradictory statements; MUST be corrected
  7. If authors can elaborate on whether there are any overlapping SNPs found in endometriosis and migraine GWAS [references 26 and 27; lines 104-105]. If yes, then are these SNPs found in the same overlapping genetic loci found in the authors’ results? Please explain appropriately.
  8. Are there any data showing that in a comorbid patient with endometriosis and migraine, which disease came first and which one later?
  9. Line 628- is the SLC35G6 gene expressed in women? Because it was mentioned that it is expressed in testis. Although the biological function is unknown, authors must mention whether the SLC35G6 gene is expressed in women or not. If the data is not found, please discuss it in the discussion section appropriately.
  10. Line 730- "Our findings further.."
  11. Lines 743-744- This conclusion is already established/published earlier. Please correct the statement appropriately as it looks like a new conclusion.
  12. In the conclusion paragraph, authors must try to explain how their contribution might help improve the current treatment of endometriosis and migraine comorbid patients; proving the clinical significance of their findings.
  13. Duplication of reference numbers should be corrected.
